# DNA Methylation Signature in Mononuclear Cells and Proinflammatory Cytokines May Define Molecular Subtypes in Sporadic Meniere Disease

**DOI:** 10.3390/biomedicines9111530

**Published:** 2021-10-25

**Authors:** Marisa Flook, Alba Escalera-Balsera, Alvaro Gallego-Martinez, Juan Manuel Espinosa-Sanchez, Ismael Aran, Andres Soto-Varela, Jose Antonio Lopez-Escamez

**Affiliations:** 1Otology & Neurotology Group CTS495, Department of Genomic Medicine, GENYO, Centre for Genomics and Oncological Research, Pfizer University of Granada Andalusian Regional Government, PTS, 18016 Granada, Spain; marisa.flook@genyo.es (M.F.); alba.escalera@genyo.es (A.E.-B.); alvaro.gallego@genyo.es (A.G.-M.); juanmanuel.espinosa@genyo.es (J.M.E.-S.); 2Sensorineural Pathology Programme, Centro de Investigación Biomédica en Red en Enfermedades Raras, CIBERER, 28029 Madrid, Spain; 3Department of Otolaryngology, Instituto de Investigación Biosanitaria ibs.Granada, Hospital Universitario Virgen de las Nieves, Universidad de Granada, 18014 Granada, Spain; 4Department of Otolaryngology, Complexo Hospitalario de Pontevedra, 36071 Pontevedra, Spain; ismaelaran2000@yahoo.com; 5Division of Otoneurology, Department of Otorhinolaryngology, Complexo Hospitalario Universitario, 15706 Santiago de Compostela, Spain; andres.soto@usc.es; 6Division of Otolaryngology, Department of Surgery, University of Granada, 18011 Granada, Spain

**Keywords:** Meniere Disease, cytokines, WGBS, hearing loss, DNA methylation

## Abstract

Meniere Disease (MD) is a multifactorial disorder of the inner ear characterized by vertigo attacks associated with sensorineural hearing loss and tinnitus with a significant heritability. Although MD has been associated with several genes, no epigenetic studies have been performed on MD. Here we performed whole-genome bisulfite sequencing in 14 MD patients and six healthy controls, with the aim of identifying an MD methylation signature and potential disease mechanisms. We observed a high number of differentially methylated CpGs (DMC) when comparing MD patients to controls (*n*= 9545), several of them in hearing loss genes, such as *PCDH15,* *ADGRV1* and *CDH23*. Bioinformatic analyses of DMCs and cis-regulatory regions predicted phenotypes related to abnormal excitatory postsynaptic currents, abnormal NMDA-mediated receptor currents and abnormal glutamate-mediated receptor currents when comparing MD to controls. Moreover, we identified various DMCs in genes previously associated with cochleovestibular phenotypes in mice. We have also found 12 undermethylated regions (UMR) that were exclusive to MD, including two UMR in an inter CpG island in the *PHB* gene. We suggest that the DNA methylation signature allows distinguishing between MD patients and controls. The enrichment analysis confirms previous findings of a chronic inflammatory process underlying MD.

## 1. Introduction

Meniere Disease (MD, MIM 156000) is a chronic disorder of the inner ear that consists of episodes of spontaneous vertigo, usually associated with low to middle-frequency sensorineural hearing loss (SNHL), tinnitus and/or aural fullness [1]. The disorder is a heterogeneous condition that usually begins in one ear with tinnitus and hearing loss, but it can involve both ears and produce bilateral symptoms in up to 40% of patients [2].

Epidemiological and familial aggregation studies suggest that MD is a multifactorial disorder with a significant heritability [3]. The condition is polygenic, including sporadic and familial cases. Several genes, such as *DTNA, FAM136A, PRKCB, DTP, and SEMA3D,* have been associated in multiplex families with autosomal dominant inheritance with incomplete penetrance [4]. In addition, six families with rare missense variants in the *OTOG* gene have been reported, supporting an autosomal recessive compound heterozygous inheritance [5].

However, the majority of patients with MD are considered sporadic and there is growing evidence to support a central role of the immune response in MD [6]. Our group found two subgroups of MD patients according to the baseline levels of IL-1β, differentiating patients with high levels of IL-1β (MDH) and patients with low levels (MDL), who in turn may have different immune response profiles to antigens or even differences in the functional status of the immune system [7].

Methylation of cytosines in the DNA strand is a stable epigenetic mechanism essential in regulating gene expression and determining the phenotype of a cell. There is numerous evidence demonstrating the nature of epigenetics to human biology and pathology. Despite extensive research in methylation, little work has been conducted examining how epigenetic changes affect gene expression in hearing. Epigenetics and, therefore, DNA methylation, could play an important role in hearing-related diseases that have no identifiable perturbation to the DNA sequence, even in those with known mutations, epigenetic modifications could be important to phenotypic differences [8,9].

Previous DNA methylation studies on hearing loss with humans were made with different technologies: Reduced Representation Bisulfite Sequencing, arrays and methylation-specific PCR [10,11,12,13]. Nevertheless, Whole Genome Bisulfite Sequencing (WGBS) is perhaps the most powerful method to interrogate the methylome as it potentially allows investigation of every 5′—C—phosphate—G—3′ (CpG) site in the genome (20–22 million CpGs are usually covered in the mappable human genome) [8]. Yizhar-Barnea et al. [14] used WGBS to obtain the first DNA methylome map of the mouse inner ear sensory epithelium, which revealed novel regulatory regions in the hearing organ.

In this study, we performed WGBS in patients with MD and healthy controls with the aim of identifying an MD methylation pattern and potential disease mechanisms.

## 2. Materials and Methods

### 2.1. Human Subjects

We included a total of 14 patients with definite MD and six healthy controls that were recruited between January and July 2019, from Spanish referral centers. Patients were diagnosed according to the diagnostic criteria of the Barany Society for MD [1]. The experimental protocols of this study were approved by the Institutional Review Board in all participating hospitals and every patient signed written informed consent. The study was carried out according to the principles of the Declaration of Helsinki revised in 2013 for investigation with humans.

### 2.2. Clinical Data

A descriptive analysis was conducted using IBM SPSS Statistics v19 (IBM Corp, Armonk, NY, USA) for all clinical data. Patients were classified according to the cytokine levels and clinical variables were compared between both groups by applying Pearson’s chi-square test for qualitative variables and Student’s *t*-test for the quantitative ones. The level of significance considered was *p*-value < 0.05.

### 2.3. DNA Extraction

DNA was extracted from peripheral blood mononuclear cells (PBMCs) using the QIAamp DNA Blood Mini Kit (Qiagen, Hilden, Germany), following the manufacturer’s protocol. DNA concentration and quality parameters were verified by Nanodrop (Thermo Fisher, Waltham, MA, USA) and Qubit (Invitrogen, Waltham, MA, USA) as previously described [15]. Additionally, DNA integrity was verified by electrophoresis in a 2% agarose gel. For WGBS the minimum parameters considered were a concentration superior to 20 ng/µL, a 260/280 ratio superior to 1.8 and no observable smearing/DNA degradation by electrophoresis.

### 2.4. WGBS Library Preparation

WGBS was carried out in 20 samples (7 MDH, 7 MDL and six healthy controls) by Macrogen (Seoul, Korea). Briefly, Accel-NGS Methyl-Seq DNA Library Kit (Zymo Research, Irvine, CA, USA) was used to prepare NGS libraries from bisulfite-converted DNA for sequencing [16]. For this, the samples were treated with bisulfite to convert the unmethylated cytosines to uracils, while retaining the methylated. This was followed by an Adaptase step that performed tailing and ligation of truncated adapters to 3′ ends. The extension and ligation steps added truncated adapters, which was followed by an indexing PCR step increasing the yield and incorporating full-length adapters for single or dual indexing. Finally, bead-based clean-ups removed oligonucleotides and small fragments.

### 2.5. WGBS Data Analysis

After sequencing on a NovaSeq 6000 system, raw sequence reads were filtered based on quality and adapter sequences were trimmed. Those sequences were mapped to the reference genome with BSMAP [17], based on the SOAP (Short Oligo Alignment Program). The only uniquely mapped reads were selected to sort and index, and PCR duplicates were removed with SAMBAMBA (v0.5.9) [18].

The reference genome used was hg19, then we decided to perform a lift over in the CpGs coordinates to the hg38 reference genome through the LiftOver tool from USCS (http://genome.ucsc.edu/, accessed on 4 June 2020). CpG sites with less coverage than 10X, more methylation than 99.9% and not appearing in all the samples were filtered. The methylation ratio of every single cytosine location was extracted from the mapping results using the methylKit R package [19]. The coverage profile results were calculated as the number of C/effective CT counts for each cytosine in CpG. The batch effect was corrected by the origin of patients, with the limma R package [20]. Each Differentially Methylated CpG (DMC) was annotated by Bedtools [21] intersected using the comprehensive gene annotation on the reference chromosome from Gencode v33 [22], this included the functional location of each gene, gene ID and strand. Besides, DMCs in promoters were annotated using the Bedtools window with the same annotation file. Promoters were defined as the 1000 bp region before the transcriptional start site.

Furthermore, for DMC, different comparisons between MDH, MDL and controls were performed and filtered out through statistical hypothesis testing using independent Student’s *t*-test for sites with a minimum 8% difference in methylation. We considered significant a False Discovery Rate (FDR) adjusted *p*-value < 0.05.

For differentially methylated regions (DMR), the calling radmeth command-line tool in the Methpipe software package was used [23]. CpG sites with more methylation than 99.9% and not appearing in all the samples were filtered. The tool takes into account the coverage for each CpG site, so coverage was used from 1X. After that, *p*-values were corrected by the *p*-values of its neighbors located at a distance of 200 from each other, using the parameter 1:200:1. We required DMRs to contain at least two DMCs, a minimum of 8% difference in methylation and a corrected *p*-value < 0.05 [24,25]. Genes and promoters were annotated using Bedtools.

### 2.6. Undermetlylated Regions

For each sample, CpG sites with coverage below 10X were filtered. The R package methylSeekR was used [26]; firstly, partially methylated domains (PMDs) were identified by a hidden Markov model and they were discarded. The following criteria were applied to identify undermethylated regions (UMRs): FDR < 5% for regions and average DNA methylation < 10%. Using multiIntersectBed from BedTools [21] overlapping UMRs in MD samples and in control samples were found, those UMRs that appear in more than the 75% of MD samples and not in control samples were selected for further analyses.

### 2.7. Inner Ear Gene Sets

DMCs from the mapped genes for each comparison (MD patients vs. controls, MDH vs. controls, and MDL vs. controls), were filtered by the following gene sets:

Sensorineural hearing loss genes retrieved from Deafness variation database (https://deafnessvariationdatabase.org/, accessed on 10 April 2021), referred from now on as HL gene set (*n* = 224); genes showing a burden of rare variants in sporadic MD referred from now on as SMD gene set (*n* = 70) [27], and differentially expressed genes (DEGs) in the mouse stria vascularis single cell RNAseq dataset [28], referred from now on as SV gene set (*n* = 217).

### 2.8. Functional Analysis

Goseq package was used with the aim of defining biological pathways, processes and functions using Gene Ontology (GO) and Kyoto Encyclopedia of Genes and Genomes (KEGG) databases [29]. To correct the bias that occurs due to different gene lengths and distinct number of CpGs per gene after filtering, a bias value for each gene was formulated considering the weight of the number of CpGs in that gene after filtering summed to the inverse of the mean of the absolute values of differential methylation (DM) for all the DMCs in the gene:$$\mathrm{bias}\;\mathrm{value}=\mathrm{nCpGs}\;+\frac{1}{\frac{\sum^{}\left|\Delta \mathrm{DM}\right|}{\mathrm{nDMC}}}$$

So, genes with more CpGs have a bigger bias and genes with DMCs with higher differential methylation would have a lower bias value.

This functional analysis was also done separating the genes based on whether they contained hypomethylated or hypermethylated DMCs, in each case the bias value was calculated for those DMCs.

The Genomic Regions Enrichment of Annotations Tool (GREAT) version 4.0.4 (http://great.stanford.edu/public/html/, accessed on 17 May 2021) was used with the output of Methpipe software package, this is DMCs and DMRs coordinates, filtered for a minimum of 8% difference in methylation and a FDR adjusted *p*-value < 0.05. Gene regulatory domain definition was set as: 5 kb upstream, 1 kb downstream and a plus distal of 1 kb [30]. Results were considered significant if both the binomial test over geometric regions and the hypergeometric test over genes produced FDR q-values below 0.05, and if binomial fold enrichment is over 2.

The findMotifsGenome functionality from HOMER Motif Analysis software was used to perform the Transcription Factor motif enrichment [31].

Lastly, a transcription factor enrichment analysis was carried out with Genecodis [32] using a hypergeometric test.

### 2.9. Visualizations

The following R packages were used for the visualizations: circlize for circus plot [33]; annotatr for the distribution of CpG islands [34]; pheatmap for the heatmap [35], which contains a hierarchical clustering with euclidean distance; VennDiagram for the Venn Diagrams [36]; and ggplot2 [37], dplyr [38], forcats [39] and ggpubr [40] for the remaining visualizations.

## 3. Results

### 3.1. Patient Clinical History

Table 1 shows the clinical features of 14 MD patients (seven MDH and seven MDL) and six healthy controls. No differences were found for any of the controlled variables (*p*-value > 0.071). Patients with MD can be classified into different clinical subgroups according to several comorbidities, such as migraine or autoimmune disorders. Most patients belong to the clinical subtype 1 of MD, independently of the level of cytokines (*p*-value = 0.401). One of the patients suffers from antiphospholipid syndrome, an autoimmune disorder (subtype 5 of MD).

### 3.2. Screening DNA Methylation in Mononuclear Cells in Sporadic Meniere Disease

WGBS was performed to compare the methylation profile in mononuclear cells of patients with MDH, MDL and controls. A total of 53,505,405 CpG sites were identified, after quality filtering for a minimum of 10X coverage per sample and methylation below 99.9%, 704,312 sites remained. We observed that differentially hypomethylated sites were more frequent along the genome (Figure 1A). We defined the distribution of the differentially methylated sites according to the functional position (promoter, intronic, intergenic, exonic, 5′UTR, 3′UTR) (Figure 1B) and we found that most sites were intronic (49.19% of hypomethylated DMCs and 3.81% of hypermethylated DMCs) and least frequently in the 5′UTR (0.02% of hypomethylated DMCs). We defined the distribution of DMCs across CpG islands and their neighboring regions and observed that CpG islands were the only region where there were more hypermethylated DMC (0.07%) than hypomethylated DMC (0.04%) (Figure 1C).

CpG methylation hierarchical clustering showed that MD patients form a different group than healthy controls and patients with MDH and MDL are also clustered separately (Figure 2A).

### 3.3. Undermethylated Regions in Meniere Disease

The UMRs (<10% average methylation) in MD and control genomes were retrieved and mapped to the different genomic regions. There was a significant difference in the number of UMR between MD patients and controls in all genomic regions (*p*-value < 0.0069). We also observed that CpG shores displayed the highest number of UMR (Figure 3).

Next, UMRs were filtered to identify which were found in at least 75% of patients with MD, but not in controls. Namely, we identified two UMRs in the inter CpG region of the *PHB* gene (Appendix A).

### 3.4. Mapping Differential Methylated Sites

We found a total of 19,055 DMCs when comparing MDH patients to healthy controls (*p*-value < 0.05) (Figure 2B, Appendix A). A total of 10,333 DMCs were uniquely found when comparing MDH to controls (*p*-value < 0.05), which were mapped to 1721 genes (Figure 2C).

For MD compared to controls, we identified 124 DMRs with two or more DMCs and 96 DMCs or DMRs that were mapped to promoter regions (Appendix A and S8). Only the *IL9RP3* gene had both mapped DMR and DMC in the promoter region.

Comparing MDH to controls, we identified 144 DMRs with two or more DMCs and 106 DMCs or DMRs that were mapped to promoter regions (Appendix A and S9). Three genes *H3Y1, ACSBG1* and *IL32* had both mapped DMR and DM in the promoter region.

We identified 36 DMRs with two or more DMCs, and 38 DMCs or DMRs that were mapped to promoter regions, when comparing MDL to controls, none of which were found in the same gene (Appendix A and S10).

In all the comparisons the difference in methylation was greater than 8%.

### 3.5. Hearing Loss Gene Sets

We filtered the mapped DMCs between MD and control by three gene sets: (a) Sensorineural hearing los genes retrieved from Deafness variation database (https://deafnessvariationdatabase.org/, accessed on 10 April 2021), (b) genes showing a burden of rare variants in sporadic MD (SMD) and (c) differentially expressed genes (DEGs) according to mouse stria vascularis single cell RNAseq dataset. We found 68 DMCs (adjusted *p*-value < 0.05) that were mapped to 35 stria vascularis genes, 29 DMCs (adjusted *p*-value < 0.05) that were mapped to 12 SMD genes and 60 DMCs (adjusted *p*-value < 0.05) that were mapped to 30 hearing loss genes (Figure 4A, Appendix A). *DMXL2* was the only gene shared between the three gene sets that had differential methylation between MD and controls (Figure 4A).

We calculated the mean of all CpGs mapped to the genes in the above-mentioned gene sets. We observed that there were significant differences in all disease groups when compared to controls (*p*-value < 5.7 × 10^−7^) and that MD patients had generally lower methylation in those genes (Figure 4B). Moreover, we observed that MDH patients were less methylated in those genes than MDL patients (*p*-value < 0.006) (Figure 4B).

A list of the top 10 DMCs for each comparison and its corresponding positions with a higher difference in methylation can be found in Table 2.

### 3.6. Functional Analysis

A functional analysis was carried out with goseq using the DMCs’ mapped genes which were found in the different comparisons. We found that the over-represented GO terms most associated with these comparisons are related to the cellular membrane and that under-represented terms are most associated with metabolic processes (Appendix A). We also observed that the terms associated with synaptic and postsynaptic membranes were only significant in the comparison between MDH and controls (Appendix A). In Table 3, the results for KEGG pathway enrichment analysis in the comparison between MD and controls when accounting for all mapped genes with DMCs can be observed. When considering only the genes with mapped hypomethylated DMCs, no results were found for the comparison between MDL and controls; for MDH compared to controls, only the neuroactive ligand-receptor interaction pathway was significant (64/145 genes, *p*-value = 8.63 × 10^−6^), and for MD compared to controls seven metabolism-related pathways were considered significant (Table 3).

No GO terms nor KEGG pathways were associated with DMRs or promoters from any of the comparisons.

GREAT was used to predict the function of cis-regulatory sites and regions, with DMCs and DMRs extracted from Methpipe. For the comparison between MD and controls, and MDL and controls, the same results based on mouse phenotype were obtained (Appendix A). We observed that most phenotypes were associated with cochlear and organ of Corti degeneration, and abnormal synaptic currents (Figure 5, Appendix A). For the comparison between MDH and controls, we also found phenotypes associated with abnormal inner ear morphology (Appendix A).

HOMER was used to search for known transcription factor binding sites in MD DMRs. A total of 23 of 440 motifs were enriched over the background in these DMRs (Table 4). The identified motifs were enriched in binding sites for transcription factors that are mostly related to immune response and inflammatory response (Appendix A). The enriched binding sites differ in the type of DNA-binding domain greatly, however, there is a predominance of homeobox domains (6/23).

## 4. Discussion

The main findings of this study are that MD patients show a different methylation profile in mononuclear cells from controls, which may be associated with an increased activated state of immune cells in MD. The present study aimed at identifying differential methylation patterns between MD patients and healthy controls to identify potential mechanisms and putative disease targets. Taken together, these results suggest the involvement of methylation in various hearing loss, sporadic MD and SV genes, as is *KCNE1* which is differentially expressed genes in the marginal cells of the mouse SV [28], and *ADGRV1* and *PCDH15* which encode for proteins forming ankle links in the stereocilia bundle [41,42,43].

A genome coverage of 15X was selected for this study, based on the work of Ziller et al., who concluded that coverage in the range of 5X to 15X was sufficient for the identification of DMRs [44].

Li et al. described the methylome status of human PBMCs and observed that there were no chromosome-specific effects and that most identified CpGs were intronic [45], which is in agreement with our findings (Figure 1).

We have identified 12 UMR exclusive to MD (Appendix A). *PHB* gene presented 2 UMR in an inter CpG islands. *PHB* encodes Prohibitin, a protein with a role in B cell receptor signaling, antigen-stimulated signaling in mast cells, T cell maturation and mitochondrial integrity [46]. Furthermore, it has been demonstrated that IL-6, a cytokine that has been observed with increased levels in MDH patients, increases PHB protein and induces *PHB* promoter activation [47]. Shi et al. have described prohibitin as an autoantigen in rheumatoid arthritis, with autoantibodies present in approximately 30% of patients [48]. Moreover, *PHB* is an important paralog of *PHB2*, which has been described to be expressed in hair cells and spiral ganglion and might be related to mitophagy in age-related hearing loss [49].

We observed a higher number of DMCs when comparing MDH to controls (*n* = 19,055) than when comparing MDL to controls (*n* = 2635). This suggests that the methylation changes are prominent in patients with higher levels of cytokines, supporting the hypothesis of a different functional state in these patients.

KEGG analysis indicated that the genes with hypomethylated DMCs in MD were from metabolism-related pathways (Table 3), such as ascorbate and aldarate metabolism, pentose and glucuronate interconversions, and starch and sucrose metabolism, which take part in carbohydrate metabolism. Various studies have described how immune cell function and fate are affected by metabolic pathway choices [50]. These pathways are directly linked to the pentose phosphate pathway (PPP) and glycolysis. Glycolysis has been recognized as a metabolism hallmark of various immune cells activation [51]. This allows the channeling of glucose-6-phosphate into the PPP, which consequently leads to the biosynthesis of amino acids required for cytokine production [52]. Interestingly, the gene with the biggest DMR in all comparisons (six DMCs in MD, six DMCs in MDH and five DMCs in MDL) (Appendix A) is *ME1* gene. *ME1* gene codes an NADP-dependent malic enzyme that links the glycolytic and citric acid cycles, which further supports a metabolic state associated with immune cell activation.

Cytochrome P450 metabolism was also associated with MD, due to differentially hypomethylated sites in *CYP2B6, CYP2C19, CYP2C9, CYP2F1, CYP3A5* genes (Table 3). It is believed that all members of *CYP2* and *CYP3* families might have a role in eicosanoid synthesis and degradation, which have various roles in inflammation, both pro-inflammatory and in resolution [53].

Retinol metabolism leads to the production of its active metabolite—retinoic acid (RA)—having a pivotal role in immune responses, which can be tolerogenic, by induction of Tregs, or pro-inflammatory, by Th1, Th2, and Th17 response, depending on the microenvironment [54]. It is believed that RA concentrations are controlled by the expression of ALDH1a enzymes, for which the coding gene *ALDH1A1, ALDH1A2 and ALDH1A3* are hypomethylated in MD (Table 3), and by CYP26 enzymes [54]. An imbalance in these enzymes could result in pathology. In fact, abnormal RA metabolism has been observed in ulcerative colitis patients and Crohn’s Disease [55]. Rampal et al. found that patients with active ulcerative colitis had elevated levels of RA in the inflamed mucosa, which was positively correlated with IL-17 and IFNγ levels [56]. Morita et al. observed that ILCregs from patients with chronic rhinosinusitis can be generated from ILC2 by RA, but are sparse in non-inflamed sinus tissue [57]. On the other hand, Ono et al. have described the expression of Cyp26b1 and Aldh1a3 in murine developing vestibular organs and demonstrated that reduced RA signaling is required for the regional formation of the striolar/central zones [58].

*IL32* gene was found to have a DMR (DM = −0.35) and a DMC (DM = −0.41) in the promoter region when comparing MDH patients to controls (Appendix A), which is suggestive of an increased expression of *IL32* in MDH patients. IL-32 is increased by T-cell or NK cell activation, inducing the production of TNFα, IL-8, IL-1β and macrophage inflammatory protein 2 by myeloid cells [59]. On the other hand, IL-32 can be induced by IL-1β, which is increased in MDH patients. Moreover, increased serum levels of IL-32 have been associated with various auto-immune and allergic diseases, namely type 2 diabetes, asthma, allergic rhinitis, and systemic lupus erythematosus [59]. Specifically, Meyer et al. have described reduced *IL32* methylation in T CD4+ cells of juvenile idiopathic arthritis patients [60].

Taken together, these results suggest an activated state of dendritic cells—producers of retinoic acid—which could be inducing a pro-inflammatory response by T cells, as described in other autoimmune diseases [54].

Inner ear hair cells have a mechanosensory capacity due to their stereocilia, and they transduce mechanical force generated by sound waves (cochlea) or head movement (vestibular system) into electrical signals. These cells possess a hair bundle, a morphological specialization in the apical surface, formed by F-actin based stereocilia arranged in a staircase manner. Near the tip of stereocilia, the sensory mechanoelectrical transduction (MET) channel can be found, which opens when stereocilia deflection occurs [61].

We observed that there is a high number of DMCs in genes that have been previously associated with hearing loss (https://deafnessvariationdatabase.org/, accessed on 10 April 2021), sporadic MD [27], and that show a differential expression in mouse stria vascularis [28] (Appendix A). Moreover, various of these genes were also associated with the mouse phenotypes predicted by the differentially methylated DMCs and DMRs found between patients and controls, such as *PCDH15, ADGRV1, MYO15A, CLIC5, SLC26A5, LRTOMT* and *ILDR1* genes (Appendix A). *PCDH15* gene can be found in the hearing loss and sporadic MD gene sets and was associated with 15 different predicted mouse phenotypes, of which cochlear degeneration, organ of Corti degeneration and impaired swimming were found in the three comparisons. *FOXI1, NF2, MYO15A* and *CLIC5* genes, were among others associated with swimming impairment and hearing loss (Appendix A).

We found various hypomethylated DMCs in the *PCDH15* and *CDH23* genes when comparing patients to controls. *PCDH15* and *CDH23* encode proteins forming the tip-links that connect adjacent stereocilia of mechanosensitive hair cells and its disruption eliminates transduction currents [41]. *PCDH15* has various splice variants that could result in different strength and stability tip links with cadherin 23. Bouzid et al. have described that the CpG islands in *CDH23* are 3.27-fold more methylated in women with age-related hearing loss than in healthy age-matched controls and were significantly related to an increased risk of age-related hearing loss [62].

Ankle links have been described in the vestibular and auditory hair bundles and are concentrated in the region just above the insertion of the stereocilia in the cuticular plate. It is believed that USH2A and ADGRV1 compose these links [42]. Moreover, ADGRV1 is also a component of vestibular tip links. We observed differences in the methylation of these genes for all comparisons.

In vestibular hair bundles, six proteins account for more than 80% of all actin-associated molecules remaining after actin—PLS1 (plastin 1), FSCN2 (fascin 2), RDX (radixin), MYO6 (myosin 6), XIRP2 (xin actin-binding repeat-containing protein 2) and CLIC5 (chloride intracellular channel protein 5) [63]. Three of these proteins’ corresponding genes were found differentially methylated for all comparisons (Appendix A)—*RDX* and *MYO6* genes show differentially hypomethylated sites and *CLIC5* has a differentially hypermethylated site.

MET channels are localized near the lower end of the tip links and are formed by proteins encoded by *TMC1, TMC2, TMHS and TMIE*. Transmembrane channel-like protein 1 and transmembrane channel-like protein 2, encoded respectively by *TMC1* and *TMC2* have been proposed to be pore-forming of MET channels [61]. *TMC2* gene was found differentially methylated in all comparisons of MD patients to controls (Appendix A).

Spiral ganglion neurons (SGNs) connect cochlear hair cells to the cochlear nucleus in the brainstem. SGNs can be divided into type I SNGs (~95%) and type II SGNs (~5%) and form ribbon-type synapses with inner and outer hair cells, respectively, and both are excited by glutamate [64]. GREAT analysis associated phenotypes related to abnormal excitatory postsynaptic currents, abnormal NMDA-mediated receptor currents and abnormal glutamate-mediated receptor currents (Figure 5, Appendix A) to the differentially methylated sites and regions from all comparisons. Furthermore, GO analysis for the comparison of MDH to controls identified various synaptic terms (Appendix A). Synaptic loss has been previously described when SGN peripheral termini are overexposed to glutamate, causing its swelling and damage [64]. NMDA receptors biophysical characteristics contribute to synaptic dynamics in the lower auditory pathway [65]. In the inner ear, NMDA receptors have been linked to tinnitus, by overactivation of these receptors and calcium influx, leading to aberrant excitation of the auditory nerve [66].

Together these results show methylation changes in genes involved in the stereocilia formation and mechanical transduction. The epigenetic changes observed in these genes could explain incomplete penetrance and variable expressivity found in MD symptoms even in some families, as it has been previously described in other disorders, such as Non-syndromic cleft lip and/or palate, Helsmoortel-van der Aa syndrome or Phelan-McDermid syndrome [27,67,68,69,70].

Generally, hypermethylation of DNA at a DMR could negatively influence the capacity of a given transcription factor to bind to its recognition motif [71]. So, enriched motifs with CpG dinucleotides within the consensus binding motifs, such as ZFX and NPAS2, that are modified could block the binding of the transcription factors [71,72]. For motifs, such as SOX10, which do not present CpG dinucleotides in the consensus binding element suggest that increased methylation within the DMR containing these motifs may be a result of loss of transcription factor binding [71]. Our study has some limitations. First, the number of individuals included in the study is small, however, this work will serve as a pilot study of methylation in MD, which should be further validated with an increased number of patients. Second, we could not generate RNAseq data from the same individuals, which would facilitate the functional interpretation of methylation status in some of the genes.

## 5. Conclusions

We conclude that the methylation pattern allows distinguishing the MD patients from controls, as well as MD patients with high or low levels of cytokines. Moreover, the differences in methylation are higher between MDH and controls than between MDL and controls. Taken together, the enrichment analysis supports our previous findings of a chronic inflammatory process underlying MD. Furthermore, we found various DMCs in genes that have been previously associated with cochleovestibular phenotypes in mice.

## Figures and Tables

**Figure 1 biomedicines-09-01530-f001:**
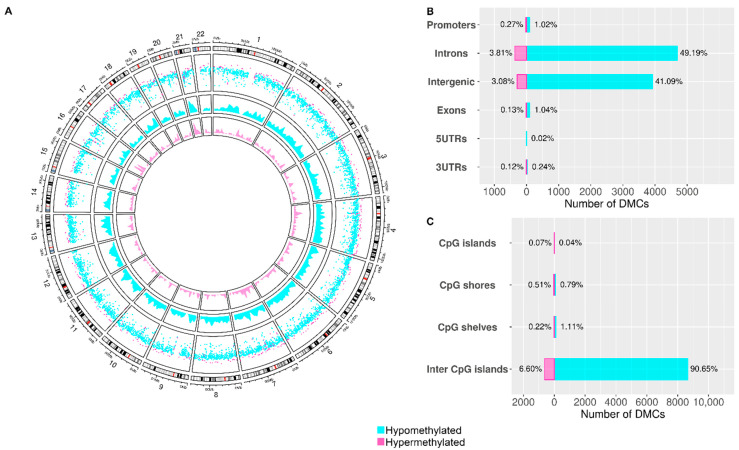
Genome wide methylation differences between Meniere Disease (MD) and controls. (**A**) circos plot representative of the distribution of the hypomethylated DMCs (blue) and hypermethylated DMCs (pink) per chromosome. (**B**) bar plot indicating the percentages of hypomethylated DMCs (blue) and hypermethylated DMCs (pink) according to the genetic region (intronic, exonic, intergenic, 5′ UTR, 3′ UTR, promoter); (**C**) bar plot indicating the percentages of hypomethylated DMCs (blue) and hypermethylated DMCs (pink) per island features (inter CpG island, CpG shore, CpG shelves, CpG island).

**Figure 2 biomedicines-09-01530-f002:**
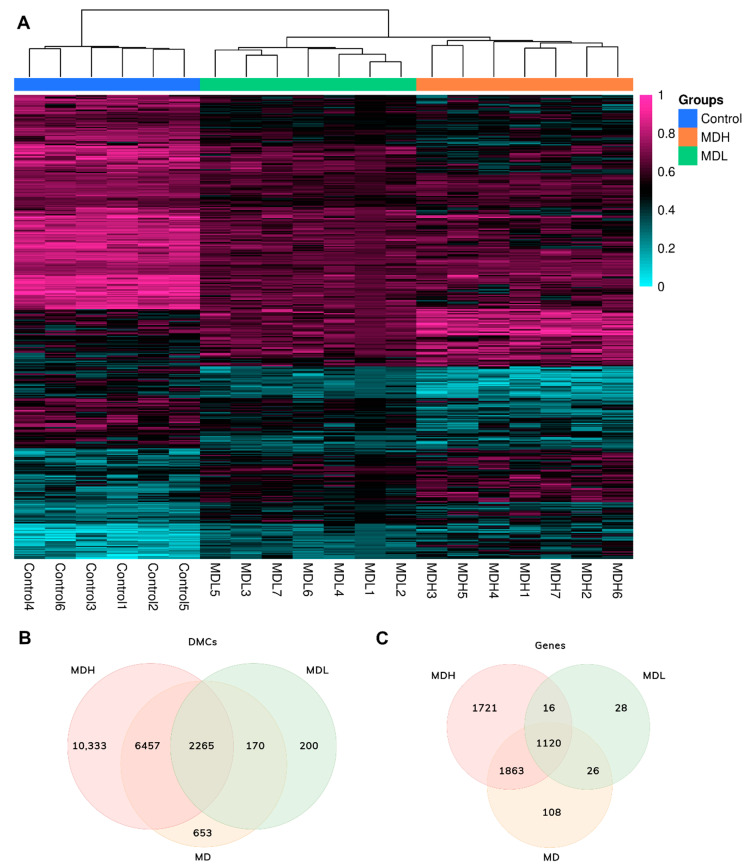
Differentially methylated CpGs and genes in Meniere Disease patients and controls. (**A**) The rows are the methylation ratio (from 0 to 1) of each position and the columns are representative of each individual, which are classified by control, MDH (Meniere Disease High) and MDL (Meniere Disease Low). The positions represented were those with a higher absolute value of the difference in methylation: 150 DMCs (differentially methylated CpGs) that show the difference between Meniere Disease and control (Appendix A), 250 DMCs for MDH (Appendix A) and 100 DMCs for MDL (Appendix A); (**B**) Venn diagram of DMCs (DM > 8%) comparing Meniere Disease to controls (MD), Meniere disease with high levels of cytokines to controls (MDH) and Meniere Disease with low levels of cytokines to controls (MDL); (**C**) Venn diagram of annotated genes from DMC analysis comparing Meniere Disease to controls (MD), Meniere disease with high levels of cytokines to controls (MDH) and Meniere Disease with low levels of cytokines to controls (MDL).

**Figure 3 biomedicines-09-01530-f003:**
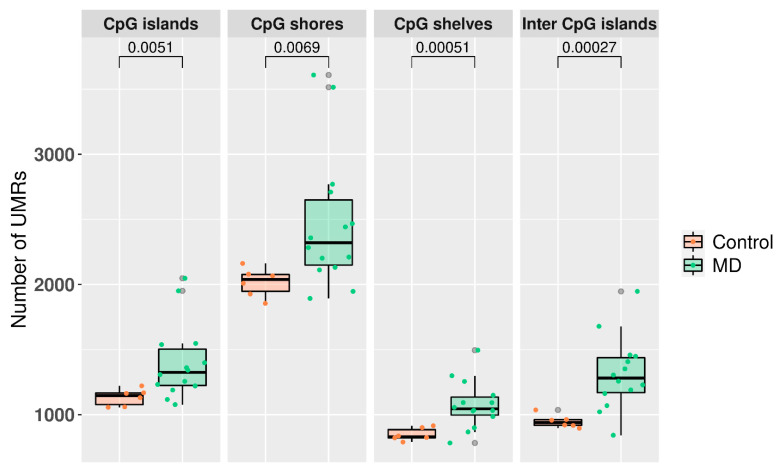
Boxplot representing the genomic region distribution of undermethylated regions (UMR) in MD and controls. Student’s *t*-test was used to calculate the *p*-value in each comparison.

**Figure 4 biomedicines-09-01530-f004:**
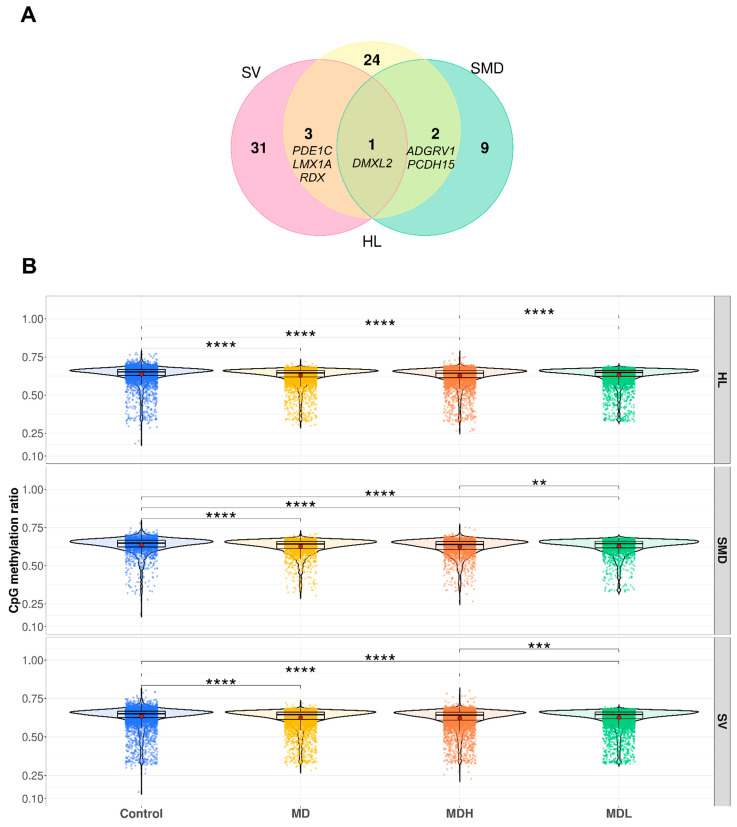
Differentially methylated CpGs present in inner ear gene sets. (**A**) Venn diagram representative of the number of genes in the Stria Vascularis (SV), HL (hearing loss) and Sporadic Meniere Disease (SMD) gene sets, which have DMC when comparing MD patients to controls. (**B**) boxplot representing the mean methylation of each CpG in the genes present in the Stria Vascularis (SV), HL (hearing loss) and Sporadic Meniere Disease (SMD) gene sets in controls (blue), all Meniere disease patients (yellow), Meniere disease patients with high cytokines (MDH) (orange) and Meniere disease patients with low cytokines (MDL) (green). Independent Student’s *t*-test was used to calculate the *p*-value in each comparison (Appendix A). **—*p*-value < 0.01; ***—*p*-value < 0.001; ****—*p*-value < 0.0001.

**Figure 5 biomedicines-09-01530-f005:**
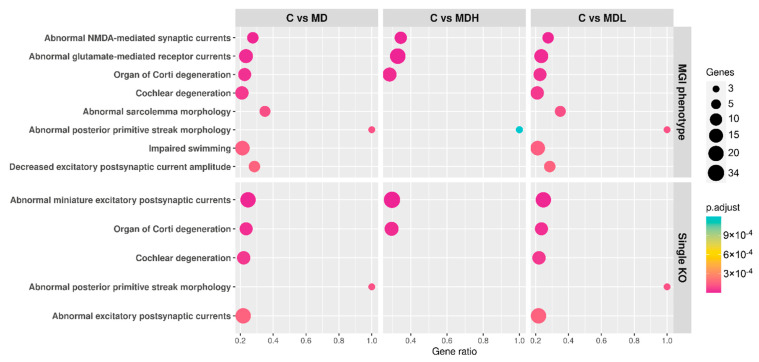
Mouse phenotype associated with differentially methylated CpGs and regions. Dot-plot representing the gene ratio and adjusted *p*-value associated to each mouse phenotype, retrieved from comparing Meniere disease patients to controls, Meniere disease patients with high cytokines to controls, and Meniere disease patients with low cytokines to controls in GREAT web tool.

**Table 1 biomedicines-09-01530-t001:** Clinical and demographic variables were assessed in patients with Meniere Disease with high levels of IL-1β (MDH), Meniere Disease with low levels of IL-1β (MDL) and controls.

Variable	MDH (*n* = 7)	MDL (*n* = 7)	Controls (*n* = 6)	*p*-Value
Age (mean ± SD)	59.6 ± 11.4	46.0 ± 11.8	51.2 ± 13.8	0.11
Age of onset (mean ± SD)	50.2 ± 9.9	37.6 ± 12.4	-	0.07
Sex (% female)	42.9 (3)	71.4 (5)	33.3 (2)	0.35
Laterality (% unilateral)	28.6 (2)	57.1 (4)	-	0.39
Ear Family History (%)	0 (0)	14.3 (1)	-	0.36
Migraine (%)	14.3 (1)	14.3 (1)	-	0.91
History of autoimmune disease (%)	0 (0)	14.3 (1)	-	0.30
Clinical Subtype (%)				
1 (no autoimmune disorder)	83.3 (5)	71.4 (5)	-	0.40
2 (delayed MD)	0 (0)	14.3 (1)	-	
3 (familial history of MD)	0 (0)	0 (0)	-	
4 (MD and migraine)	16.7 (1)	0 (0)	-	
5 (MD with autoimmune disorder)	0 (0)	14.3 (1)	-	

**Table 2 biomedicines-09-01530-t002:** Top 10 DMCs ranked according to ∆Mean value found in hearing loss (HL), sporadic Meniere disease (SMD) and stria vascularis (SV) gene sets when comparing MD patients to controls. *p*-value is adjusted by FDR. ∆Mean—difference in methylation between MD patients and controls.

Gene Set	Gene	Protein Activity or Function/Location	Position	∆Mean	*p*-Value
HL	*MSRB3*	Reduction of methionine sulfoxide to methionine	chr12:65397684	−0.20	5.77 × 10^−3^
	*PTPRQ*	Plasma membrane tyrosine phosphatase receptor	chr12:80550423	−0.18	6.49 × 10^−3^
	*ADGRV1*	G-protein coupled receptor, binds calcium	chr5:90721360	−0.15	5.73 × 10^−3^
	*ADGRV1*	G-protein coupled receptor, binds calcium	chr5:90665789	−0.15	2.64 × 10^−2^
	*MSRB3*	Reduction of methionine sulfoxide to methionine	chr12:65440113	−0.15	9.32 × 10^−3^
	*CACNA1D*	Voltage-dependent calcium channel	chr3:53684153	−0.14	9.09 × 10^−4^
	*USH2A*	Usherin—maintenance of the hair bundle ankle formation	chr1:215677582	−0.13	6.81 × 10^−5^
	*LMX1A*	Transcriptional activator	chr1:165321950	−0.13	2.04 × 10^−4^
	*PCDH15*	Membrane protein that mediates calcium-dependent cell-cell adhesion	chr10:54924915	−0.12	1.83 × 10^−4^
	*ATP2B2*	Intracellular calcium homeostasis	chr3:10545443	−0.12	1.85 × 10^−5^
SMD	*ADGRV1*	G-protein coupled receptor, binds calcium	chr5:90721360	−0.15	5.73 × 10^−3^
	*ADGRV1*	G-protein coupled receptor, binds calcium	chr5:90665789	−0.15	2.64 × 10^−2^
	*ADAM12*	Cell-cell and cell-matrix interactions	chr10:126355102	−0.13	3.83 × 10^−4^
	*PCDH15*	Membrane protein that mediates calcium-dependent cell-cell adhesion	chr10:54924915	−0.12	1.83 × 10^−4^
	*TPTE*	Signal transduction	chr21:10561174	−0.12	1.78 × 10^−2^
	*MPDZ*	AMPAR potentiation and synaptic plasticity in excitatory synapses	chr9:13106557	−0.10	8.14 × 10^−3^
	*PCDH15*	Membrane protein that mediates calcium-dependent cell-cell adhesion	chr10:54280633	−0.10	2.93 × 10^−4^
	*CFTR*	Chloride channel	chr7:117360906	−0.10	2.56 × 10^−4^
	*ATM*	Cell cycle checkpoint kinase	chr11:108237615	0.10	1.29 × 10^−2^
	*PCDH15*	Membrane protein that mediates calcium-dependent cell-cell adhesion	chr10:55026981	−0.10	5.23 × 10^−3^
SV	*ROBO2*	Axon guidance and cell migration	chr3:76840338	0.25	9.46 × 10^−3^
	*ROBO2*	Axon guidance and cell migration	chr3:76611689	−0.20	2.71 × 10^−4^
	*NFKB1*	Pleiotropic transcription factor	chr4:102589956	0.19	4.21 × 10^−2^
	*DLC1*	Regulation of small GTP-binding proteins	chr8:13480274	0.16	1.79 × 10^−4^
	*BMPR1B*	Transmembrane serine/threonine kinases receptor	chr4:95037875	−0.16	3.75 × 10^−2^
	*DLC1*	Regulation of small GTP-binding proteins	chr8:13446699	−0.16	8.60 × 10^−5^
	*ROBO1*	Mediates cellular responses to molecular guidance cues	chr3:79605304	−0.16	3.69 × 10^−3^
	*DLC1*	Regulation of small GTP-binding proteins	chr8:13522539	−0.15	1.82 × 10^−2^
	*PARD3*	Asymmetrical cell division and cell polarization processes	chr10:34349620	−0.14	3.23 × 10^−3^
	*ROBO1*	Mediates cellular responses to molecular guidance cues	chr3:79389106	−0.14	4.67 × 10^−2^

**Table 3 biomedicines-09-01530-t003:** Significative KEGG terms (adjusted *p*-value < 0.05) for mapped DMCs comparing MD patients to controls. nDMInCat—number of genes with DMCs; nInCat—number of genes in the category; *p*, adjust—adjusted *p*-value.

DMC	Term	Category	nDMInCat	nInCat	Ratio	*p*, Adjust	Genes
All	Retinol metabolism	830	16	40	0.40	2.03 × 10^−2^	*ALDH1A1, ALDH1A2, CYP2B6, CYP2C19, CYP2C9, CYP3A5, LRAT, UGT1A10, UGT1A3, UGT1A4, UGT1A5, UGT1A6, UGT1A7, UGT1A8, UGT1A9, UGT2B4*
	Metabolism of xenobiotics by cytochrome P450	980	15	43	0.35	2.97 × 10^−2^	*ALDH1A3, CYP2B6, CYP2C19, CYP2C9, CYP2F1, CYP3A5, UGT1A10, UGT1A3, UGT1A4, UGT1A5, UGT1A6, UGT1A7, UGT1A8, UGT1A9, UGT2B4*
Hypomethylated	Retinol metabolism	830	16	40	0.40	6.20 × 10^−3^	*ALDH1A1, ALDH1A2, CYP2B6, CYP2C19, CYP2C9, CYP3A5, LRAT, UGT1A10, UGT1A3, UGT1A4, UGT1A5, UGT1A6, UGT1A7, UGT1A8, UGT1A9, UGT2B4*
	Metabolism of xenobiotics by cytochrome P450	980	15	43	0.35	9.38 × 10^−3^	*ALDH1A3, CYP2B6, CYP2C19, CYP2C9, CYP2F1, CYP3A5, UGT1A10, UGT1A3, UGT1A4, UGT1A5, UGT1A6, UGT1A7, UGT1A8, UGT1A9, UGT2B4*
	Drug metabolism—cytochrome P450	982	14	45	0.31	2.71 × 10^−2^	*ALDH1A3, CYP2B6, CYP2C19, CYP2C9, CYP3A5, UGT1A10, UGT1A3, UGT1A4, UGT1A5, UGT1A6, UGT1A7, UGT1A8, UGT1A9, UGT2B4*
	Ascorbate and aldarate metabolism	53	10	21	0.48	2.82 × 10^−2^	*ALDH2, UGT1A10, UGT1A3, UGT1A4, UGT1A5, UGT1A6, UGT1A7, UGT1A8, UGT1A9, UGT2B4*
	Steroid hormone biosynthesis	140	13	40	0.33	3.11 × 10^−2^	*CYP3A5, CYP7B1, HSD17B3, HSD17B6, UGT1A10, UGT1A3, UGT1A4, UGT1A5, UGT1A6, UGT1A7, UGT1A8, UGT1A9, UGT2B4*
	Pentose and glucuronate interconversions	40	10	22	0.45	3.11 × 10^−2^	*ALDH2, UGT1A10, UGT1A3, UGT1A4, UGT1A5, UGT1A6, UGT1A7, UGT1A8, UGT1A9, UGT2B4*
	Starch and sucrose metabolism	500	14	39	0.36	3.11 × 10^−2^	*ENPP1, ENPP3, HK1, MGAM, SI, UGT1A10, UGT1A3, UGT1A4, UGT1A5, UGT1A6, UGT1A7, UGT1A8, UGT1A9, UGT2B4*

**Table 4 biomedicines-09-01530-t004:** Enriched transcription factor motifs in DMRs of MD patients compared to controls according to HOMER.

Motif Name	Consensus	*p*-Value
Hoxa9 (Homeobox)	RGCAATNAAA	1.00 × 10^−4^
ZFX (Zf)	AGGCCTRG	1.00 × 10^−3^
Sox10 (HMG)	CCWTTGTYYB	1.00 × 10^−3^
ZNF711 (Zf)	AGGCCTAG	1.00 × 10^−3^
Sox6 (HMG)	CCATTGTTNY	1.00 × 10^−3^
Hoxd11 (Homeobox)	VGCCATAAAA	1.00 × 10^−3^
MYB (HTH)	GGCVGTTR	1.00 × 10^−3^
Foxa3 (Forkhead)	BSNTGTTTACWYWGN	1.00 × 10^−3^
Hoxa11 (Homeobox)	TTTTATGGCM	1.00 × 10^−2^
BMYB (HTH)	NHAACBGYYV	1.00 × 10^−2^
AMYB (HTH)	TGGCAGTTGG	1.00 × 10^−2^
Zic (Zf)	CCTGCTGAGH	1.00 × 10^−2^
NFY (CCAAT)	RGCCAATSRG	1.00 × 10^−2^
Foxo3 (Forkhead)	DGTAAACA	1.00 × 10^−2^
Sox15 (HMG)	RAACAATGGN	1.00 × 10^−2^
NPAS2 (bHLH)	KCCACGTGAC	1.00 × 10^−2^
Hoxd10 (Homeobox)	GGCMATGAAA	1.00 × 10^−2^
Bcl6 (Zf)	NNNCTTTCCAGGAAA	1.00 × 10^−2^
STAT1 (Stat)	NATTTCCNGGAAAT	1.00 × 10^−2^
Hoxa13 (Homeobox)	CYHATAAAAN	1.00 × 10^−2^
CDX4 (Homeobox)	NGYCATAAAWCH	1.00 × 10^−2^
TFE3 (bHLH)	GTCACGTGACYV	1.00 × 10^−2^
Smad4 (MAD)	VBSYGTCTGG	1.00 × 10^−2^

## Data Availability

The dataset supporting the conclusions of this article is available in the European Nucleotide Archive (ENA; hosted by the EBI) under the project ID PRJEB45377 and can be accessed via the following link https://www.ebi.ac.uk/ena/browser/view/PRJEB45377, 31 May 2021.

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
