# Peer review of "DNA Methylation Signature in Mononuclear Cells and Proinflammatory Cytokines May Define Molecular Subtypes in Sporadic Meniere Disease"

_biomedicines, 2021, doi:10.3390/biomedicines9111530_

Round 1

Reviewer 1 Report

Congratulations on a very interesting manuscript and conclusions. I have no objectives and questions to the authors. References are adequate and up to date. Manuscript is very logically written.  

Author Response

Thank you for your review and comments.

Reviewer 2 Report

Flook and colleagues analyzed the DNA methylation in different subjects by whole genome bisulfite sequencing (WGBS): 7 with MDH (MD patients with high cytokines levels), 7 with MDL (MD patients with low cytokines levels), and 6 healthy controls. Three gene sets (SNHI hearing genes [HL], sporadic MD genes [SMD], and differentially expressed genes in the mouse SV single cell RNAseq [SV]) were applied for mapping of the DMCs(differentially methylated CpG) identified. The results showed significant difference in the number of UMR (undermethylated regions) between MD patients and healthy controls. A total of 30, 12, and 35 genes were mapped to the HL, SMD, and SV gene sets, respectively. Pathway analysis of the mapped genes showed association with cochlear and organ of Corti degeneration, abnormal synaptic currents, and abnormal inner ear morphology.

In general, this is a well-organized and well-written article. The DNA methylation was compared between MD patients and healthy controls. The authors demonstrated different methylation profile among the groups, indicating different immune activated states. These findings are of clinical importance, implying possible inflammatory process in the mechanism of MD and giving the hint of MD pathogenesis. However, the number of study population is limited.

Comments:

  1. How did “healthy controls” recruit in this cohort? Since the phenotype of MD may be late-onset, how did you define “healthy controls” in the cohort?
  2. In table 1, what do “Ear Family History” refer to? What specific disease did the family member of the subject have in MDL group?
  3. In Result parts (pages 5,7,9,11), “figure 2A-2C” in text should be “figure 3A-3C”, “figure 3A-3B” in text should be “figure 4A-4B”, and “figure 4” in text should be “figure 5”.

Author Response

Thank you for your time to review our study. We appreciate your comments and questions.

Comments:

1. How did “healthy controls” recruit in this cohort? Since the phenotype of MD may be late-onset, how did you define “healthy controls” in the cohort?

We selected individuals without history of any hearing or vestibular disorder. Volunteers were asked for any audiological or vestibular symptom (tinnitus, hearing loss, dizziness, imbalance, and ear fullness) before recruitment. The selected controls were 25, 48, 53 59 and 61 years old at the time of recruitment, to match the age of MD patients.

2. In table 1, what do “Ear Family History” refer to? What specific disease did the family member of the subject have in MDL group?

“Ear Family History” refers to having a known family member that suffers from any otologic disease/symptoms. The individual in the MDL group has a first cousin with hearing loss due to antibiotic treatment.

2. In Result parts (pages 5,7,9,11), “figure 2A-2C” in text should be “figure 3A-3C”, “figure 3A-3B” in text should be “figure 4A-4B”, and “figure 4” in text should be “figure 5”.

We have modified and corrected the figure numbers in the following sections:  Screening DNA methylation in mononuclear cells in sporadic Meniere Disease; Mapping differential methylated sites; Hearing loss gene sets; and Functional Analysis.

Reviewer 3 Report

The manuscript entitled “DNA Methylation signature in mononuclear cells and proinflammatory cytokines may define molecular subtypes in sporadic Meniere Disease” has performed whole genome bisulfite sequencing in 14 MD patients and 6 healthy controls, to identify a MD methylation signature and potential mechanisms in sporadic Meniere Disease. The experiment was well designed and analyzed. They found that the DNA methylation signature can be used to select MD patients. They also confirmed a chronic inflammatory process in MD.

The work is interesting, and the manuscript is well written. I agree with the authors’ conclusion.

Author Response

Thank you for your review and comments